# Binding Zinc and Oxo-Vanadium Insulin-Mimetic Complexes to Phosphatase Enzymes: Structure, Electronics and Implications

**DOI:** 10.3390/molecules30071469

**Published:** 2025-03-26

**Authors:** Victor V. Volkov, Carole C. Perry, Riccardo Chelli

**Affiliations:** 1Independent Researcher, Bereozovaya 2a, Konstantinovo 140207, Moscow Region, Russia; 2Interdisciplinary Biomedical Research Centre, School of Science and Technology, Nottingham Trent University, Clifton Lane, Nottingham NG11 8NS, UK; carole.perry@ntu.ac.uk; 3Dipartimento di Chimica, Università di Firenze, Via della Lastruccia 3, I-50019 Sesto Fiorentino, Italy

**Keywords:** QM/MM, MD simulation, DFT, infrared, Raman, UV–VIS, circular dichroism, diabetes

## Abstract

We explore the structural and electronic properties of representative insulin-mimetic oxovanadium and zinc complexes as computed in vacuum, in water clusters and upon binding to PTEN and PTP1B phosphatases. Albeit diverse, the enzymes’ active sites represent evolutionary variant choices of the same type of biochemistry. Though different in respect to covalency and the orbital nature of bonding, theory predicts comparable ionic radii, bond lengths and square pyramidal coordination for the considered vanadyl and zinc systems when in an aqueous environment. Employing docking, DFT and quantum mechanics/molecular mechanics methods, we address possible polar interactions in the protein environments and compute infrared/Raman modes and optical electronic properties, which may be suitable for the structural analysis of the specific chemical moieties in binding studies. Accounting for how protein embedding may alter the electronic states of metal centres, we discuss artificial intelligence-assisted protein field engineering to assist biomedical and quantum information applications.

## 1. Introduction

Metal cations and their complexes have been reported as possible medical remedies since ancient times. The case of cisplatin [1] can perhaps be considered the beginning of a systematic approach to exploring how oxidation state, coordination geometry and ligand nature determine the chemistry of biomolecular interactions with medical implications [2].

Diabetes mellitus is a family of endocrine diseases characterized by high blood sugar levels. Its main forms, known as type 1 and type 2, are due to insulin insufficiency (arising from autoimmune β-cell destruction) and to the decay of β-cell insulin secretion against a background of insulin resistance that involves insulin receptor malfunction, respectively. In the first case, treatment relies on insulin injection. The treatment of patients with type 2 diabetes involves biochemotherapies, employing insulin secretion promoters, the suppression of glucose absorption, and reduction of insulin resistance [3]. Improvements in the efficiency of such treatments rely on a better understanding of type 2 diabetes biochemistry and on devising pharmacophores to compensate for diabetes signal transduction pathways with better intelligence.

The positive effects of using metal ions for the treatment of diabetes were reported in 1899, prior to the discovery of insulin by Lyonnet and coworkers [4], who observed lowered glucose levels when using sodium metavanadate [4]. In 1971, an ammonium metavanadate solution injected intraperitoneally was reported to affect glucose metabolism in rats [5]. Later studies have discussed vanadium bioactivity as stemming from the role of vanadate, $H_{2}VO_{4}^{-}$, in the regulation of phosphate-dependent processes, such as metabolic pathways relying on phosphatases, such as tyrosine phosphatase (PTP1B), and kinases [6,7]. From now on, vanadium is abbreviated as V. Reports on the significance of ligands in V bioactivity include oxovanadium complexes with maltolate, VO(mal)_2_, and picolinate, VO(pic)_2_, which were reported as effective antidiabetic remedies [8,9].

Zinc (Zn) cations came under focus as a possible antidiabetic agent in 1980, when ZnCl_2_ was reported to stimulate lipogenesis in rat adipocytes in a manner similar to the action of insulin [10], aiding both types of diabetes treatment under high doses of ZnCl_2_ [11,12]. Since then, the mechanisms of the insulin-mimetic action of Zn(II) have been examined with respect to glucose oxidation and lipolysis stimulation [13], glucose transport and glycogen synthesis [14], as well as the inhibition of endogenous glycogen synthase kinase-3β [15]. As a result of these studies, the maltolato-Zn(II) complex, Zn(mal)_2_, has been found to be potent in lowering blood glucose levels in the treatment of type 2 diabetic animals [16]. The ability of Zn compounds to improve glucose transport, glycogen synthesis and lipogenesis, to inhibit gluconeogenesis and lipolysis, and to modulate key elements of the insulin signalling pathway [17,18] has been suggested to contribute to the improvement of hyperglycemia and glucose homeostasis in diabetes.

Stimulated by the research of effective Zn- and V-based organometallic antidiabetic remedies, studies have been conducted towards the identification of proteins involved in their interactions, the speciation of active metal states when complexes exchange ligands with bioactive species such as tartrate and citrate [19,20] and the characterization, pharmacodynamics and biocompatibility that depend on the nature of the ligands present. For example, the lower solubility and stronger gastrointestinal irritation of VO(pic)_2_ with respect to VO(mal)_2_ stimulated further studies towards better biocompatibility [19]. In this respect, it is worth noting that natural allixin, found in garlic, provides vanadyl and Zn complexes with excellent capacity to lower high blood glucose levels in animal models affected by both type 1 and type 2 diabetes [21,22].

Understanding the mechanisms of antidiabetic activities concerns the identification of which enzymes are affected by Zn and V complexes. As previously mentioned, early studies suggested that V systems affect metabolic pathways relying on phosphatases, such as PTP1B, and kinases [6,7,23]. Comparatively, it has been reported that Zn^2+^ cations induce the degradation of phosphatase and tensin homologues (PTEN) [24]. In addition, the overexpression of PTEN in mice impairs the insulin signalling pathway, leading to insulin resistance [25]. Based on these observations, it has been proposed that Zn(alx)_2_ and Zn(tanm)_2_ affect PTEN and activate PI3K-Akt/PKB signalling [26].

Both vanadyl and Zn^2+^ ions, in place of insulin, induce insulin-mimetic effects with regard to both the incorporation of glucose in rat adipocytes and the inhibition of free fatty acids released from their adipocytes [27,28]. These ions have been found to simultaneously act on multiple sites in adipocytes, thus suggesting that their action should be described as an “ensemble mechanism” [29]. However, the critical action sites for V and Zn ionic species differ slightly from each other. For example, it was observed that VO(alx)_2_ targets IRβ and/or PTPase [30], while Zn(alx)_2_ and Zn(tanm)_2_ target PTEN [31].

The complexity of the discussed “ensemble mechanism” underlying the antidiabetic actions of Zn and V complexes concerns the interactions of the complexes with different enzymes, but, often, they belong to the phosphatase family, which is substrate-specific and exhibits sequence homology at the catalytic site [32]. Considering the diversity of this family as a promising palette for artificial intelligence training, we may adopt PTP1B and PTEN as archetypal phosphatase examples for introductory investigations into the interactions of V and Zn complexes with such enzymes.

PTP1B is an enzyme involved in the dephosphorylation of proteins with tyrosine residues, and hence, it plays an essential role in many signal transduction pathways [33,34,35]. It is known that the enzyme acts as a negative regulator of insulin signaling [36], and hence, it could be a pharmacological target for the treatment of type 2 diabetes and obesity [37,38,39]. PTP1B is composed of three domains: the catalytic domain (30–278) at the N-terminal, the regulatory domain (278–401) and the C-terminal (401–435), which is responsible for membrane binding [40]. The active site of the enzyme is located in a relatively shallow pocket: the site contains a P-loop with Cys215 and Arg221 residues, which are crucial for dephosphorylation processes [41,42]. The P-loop is a major binding site for orthovanadate and many other inhibitors [43,44]. It is surrounded by four structural elements: the WPD loop (Trp179-Asp181), the Q-loop (Arg257-Thr263), the pTyr-loop (Asn42-Val49) and the E-loop (Leu110-Gln123). These structures are responsible for ligand binding and substrate recognition and take part in catalytic functions [45]. Crystallographic studies have indicated that PTP1B exists in “open” and “closed” conformations, where the WPD loop moves out or inside to form an open or closed binding pocket, respectively [41,42,46,47].

PTEN is a critical negative regulator of insulin signaling through its role in the dephosphorylation of phosphatidylinositol 3,4,5-triphosphate to phosphatidylinositol 4,5-bisphosphate in the phosphoinositide 3-kinase (PI3K) pathway [48]. The enzyme has two important domains contributing to this behaviour: a DUSP domain (residues 15–185) and a C2 domain (residues 192–353). These domains are linked with three disordered segments, which play a role in membrane binding and the allosteric regulation of the enzyme [49]. Compared to PTP1B, the pocket of PTEN has a similar depth (about 8 Å), but is about twice as wide, 5 × 11 Å [50]. The active site of PTEN is made up of three loops: the WPD loop (residues 88–98), the TI loop, which contains a Thr and Ile pair and spans residues 160–171, and the P loop, with a Cys residue (C124) that is fundamental for substrate catalysis. Both the Cys and Arg residues within this loop, C124 and R130, are completely intolerant to mutations [51].

Recently, the structural properties of anticancer V and antidiabetic Zn complexes bound to PTP1B were investigated using classical molecular dynamics simulations [52,53]. To the best of our knowledge, there are no reports concerning the interactions of candidate metal–organic inhibitors with PTEN. Considering the lack of comparative studies of the structural and electronic properties of the considered Zn and V metal–organic potential inhibitors either in solution or bound to enzymes, in the present contribution, we adopt density functional theory (DFT) calculations to “refine” the predictions of docking and describe the structure and flexibility of the selected Zn and oxovanadium metalorganic systems reported to have antidiabetic properties. A quantum-level description is particularly important in the application of the V^4+^ ion, characterized by one unpaired d-electron.

In this work, we address the structural features of the complexes, especially focusing on their binding to the considered enzymes. Since significant efforts in anti-diabetes research concern the synthesis of complexes where the ligand pyromeconic acid is the central structural element [8,9,16,19,21,22], we focus on theoretical support for such a family of molecules, where the pyromeconic acid central structural element is functionalized with side groups of increasing complexity. At the same time, we give special attention to exploring the nature of bonding while comparing coordination to Zn and V cations, since their complexes demonstrate parallel antidiabetic properties. We analyze optical electronic, infrared (IR) and Raman spectral properties, which can be used for experimental diagnostics. This investigation allows us to discuss what makes the investigated complexes similar and different as antidiabetic remedies. Finally, we address biospintronic opportunities to alter metal electronics via the engineering of protein embedding in the environment, which may be tuned using Neural Network training according to the diversity of phosphatases, the predictions of DFT and experimental/empirical knowledge.

## 2. Results and Discussion

### 2.1. Dynamics in Water Clusters

In Figure 1, we present a schematic outline of the complexes considered here. In this case, when R_2_, R_5_ and R_6_ are hydrogen atoms, M is a V^4+^ cation double-bonded to O_a_, and the ligands are pyromeconic acids or 3-hydroxy-4-pyrone, which form the bis(3-hydroxy-4-pyronato) oxovanadium(IV) complex, VO(3hp)_2_. When R_2_ is a methyl moiety, the ligand is larixinic acid (or maltol, or 3-hydroxy-2-methyl-4-pyrone); with M = Zn^2+^, the complex is bis(maltolato)zinc(II), Zn(mal)_2_. Adopting a pentyl moiety for R_2_, a methoxy group for R_5_ and a methyl group for R_6_, we represent the allixin ligand (or 3-hydroxy-5-methoxy-6-methyl-2-pentyl-4-pyrone) as follows: with oxovanadium, the ligands form bis(allixinato)oxovanadium(IV), VO(alx)_2_. The image on the right side of Figure 1 shows that in the experiment, the complexes may be present as Δ and Λ enantiomers, exhibiting no optical activity. We mainly conducted studies on the Δ enantiomer, unless otherwise indicated. Exploring the structural, dynamic and optical properties of a range of possible complexes, in this contribution, we focus our attention on the characterization of the VO(3hp)_2_, VO(alx)_2_, Zn(mal)_2_ and Zn(alx)_2_ systems, which demonstrate both structural consistency and variance, and can be sufficiently informative and useful to an experimentalist in the interpretation of a range of similar but not necessarily identical structures.

In Figure 2, we present the structural and electronic properties of the VO(3hp)_2_ and Zn(mal)_2_ complexes, simulated in water clusters using BOMD. Here, it is important to stress that BOMD trajectories have been carried out in an explicit water environment starting from minimum energy structures for the complexes in vacuum obtained by DFT. In the cases of VO(3hp)_2_ and Zn(mal)_2_, the initial structures are at the global energy minima. In the case of octahedral Zn(mal)_2_·2H_2_O, the starting geometry is either at the global energy minimum or close to it. In fact, the relative orientations of the two water molecules populate the configurational space of structures with energies that vary on the scale of a few kJ/mol. Table 1 and Table 2 report some selected bond properties for the selected systems computed by DFT, while Appendix A provide an extended data set. For example, in the case of the Zn systems, DFT predicts a tetrahedral geometry for Zn(mal)_2_, as represented by the small cyan and dark red points in Figure 2A2, which reflect the angular values of the mainframe under tetrahedral geometry. The results sampled along the BOMD trajectories, as presented in Figure 2, are introductory to address why the complexes demonstrate analogous antidiabetic tendencies, whereas the metals and their valency are not alike. According to the formal valencies, we may note that the Mulliken charge for the V^4+^ cation fluctuates around higher values than is the case for Zn^2+^ (Figure 2D1,D2). At the same time, the polarities of the coordinated oxygen atoms of VO(3hp)_2_ are nearly comparable to those in the case of the Zn(mal)_2_ complex (Figure 2B1,B2), whereas the polarities of the next coordinating carbon atoms of VO(3hp)_2_ are slightly higher than in the case of the Zn(mal)_2_ complex (Figure 2C1,C2). Here, it is interesting to compare the correlations of the Mulliken charge fluctuations in both systems. Figure 2F1,F2 show comparable metal–oxygen, oxygen–oxygen and oxygen–carbon anticorrelation patterns. Nonuniform metal–oxygen anticorrelations (for example, the M-O_3_ and M-O’_3_ covariance values are not the same) suggest non-identical interactions with proximal water at opposite sides of the complexes, which is due to insufficient averaging along the finite length of the trajectories.

What is rather interesting is that in an explicit aqueous environment, Zn(mal)_2_ departs from its global minimum tetrahedral geometry (see the corresponding characteristic angular relations, as given by the small cyan and dark red points in Figure 2A2) to attract a water molecule under apical geometry (see the green line trajectory in Figure 2A2 and Mulliken reading in Figure 2E2 (to parallel analogous data for VO(3hp)_2_ as presented in Figure 2A1,E1)). The association of water is via sigma p-electron interactions, which come at the expense of the electronic depletion of carbonyl and carboxylate moieties by the metal cations. This is well reflected if we compare ESP charges computed under DFT for tetrahedral Zn(mal)_2_ and Zn(mal)_2_·H_2_O (see Table 2).

Stimulated by the results of the BOMD trajectories for Zn systems, we conducted additional DFT studies to address the structural properties of Zn^2+^ systems under coordination with explicit water molecules. Since water association with Zn(mal)_2_ in an explicit aqueous cluster is dynamic, computing this complex in vacuum (under DFT) cannot suggest a structure similar to that of VO(3hp)_2_. Only when using ab initio simulation in explicit water do we observe the coordination change from Zn(mal)_2_ to Zn(mal)_2_·H_2_O (where there is one water molecule under apical coordination), namely the transformation from a tetrahedral to pentahedral geometry analogous to that of VO(3hp)_2_. We obtained the same result when we initiated the ab initio simulation from a DFT-optimized octahedral Zn complex with two water molecules on opposite sides of the ligand plane: Zn^2+^ loses coordination with one water molecule, keeping the second dynamically associated, and bends the ligand planes, which undergo deformation. In this respect, BOMD studies of the structural and electronic properties of the complexes in an explicit water cluster are indispensable to access structural realizations of Zn^2+^ complexes prior to and upon their initial interactions with phosphatase family members. Here, it is important to mention that a deviation from the tetrahedral geometry upon the attraction of water molecules has been observed in MD simulations of alkaline earth metal cations in water clusters [54]. Furthermore, BOMD predicts that both Zn and oxovanadium systems show not only a similarity in the angular ranges spanned by their structural realizations (blue and red trajectories in Figure 2A1,A2) but also that such angular properties are featured by analogous anticorrelations (see antidiagonal character of the trajectories).

This observed similarity, however, does not mean the complexes show exactly the same structural and electronic properties. The dynamic association of water due to p-electron interactions with Zn^2+^ is not as strong and stable as V^4+^=O double bonding with the participation of the constructive p-d electronic overlap. As a result, the green trajectory in panel A2 explores a broad angular space (compared to that in Figure 2A1), reflecting the rather unstable nature of water coordination, while the Mulliken charge on oxygen (see Figure 2E2) is predicted to show significant fluctuations due to water OH stretching dynamics and hydrogen bonding dynamics involving other neighboring water molecules.

It is worth noting that borrowing from the relatively weak nature of the apical aqueous O association (in contrast to the strong V=O bonding in an oxovanadium complex), Zn systems may demonstrate unprecedented structural flexibility upon binding to a protein. Overall, a Zn complex may vary from a distorted tetragonal prism to a distorted tetrahedron. Such structural flexibility may play a significant role upon possible post-binding speciation [19,20] to release the ligands and the metal cation—a process in which protein carboxylate moieties may be involved.

### 2.2. Structure upon Binding to Proteins

For the structures of a phosphatase and tensin homolog (PTEN) and protein tyrosine phosphatase 1B (PTP1B), we took the structural information from crystallographic studies according to the RCSB Protein Data Bank PDB ID 5BZX [55] and PDB ID 1ONZ [56]. The obtained structures were processed as we describe in the Methods section. Next, inspired by BOMD and DFT outcomes, we carried out docking predictions to find candidate arrangements of the complexes within the space of the reaction centres that may lead to enzymes inhibition. In order to accurately account for the quantum nature of the metal ions, and especially the peculiarity of the unpaired electron of V^4+^, starting from the structural poses obtained by docking results, we performed ONIOM QM/MM optimizations of the systems. In Figure 3, we report ball and stick representations of the optimized complex–enzyme configurations. In the Appendix A, we provide Gaussian log files of the optimized geometries.

QM/MM calculations predicted that V complexes undergo a strong folding distortion in order to occupy the reactive center of PTP1B. In the case of VO(3hp)_2_, binding is dominated by polar interactions with the Phe122 backbone oxygen, backbone and side group hydrogens of Arg221, side group hydrogens of Gln266 and Gln262, a side group hydrogen of C215 and a side group oxygen of Tyr46 (see Figure 3A). VO(alx)_2_ interactions with the protein account for polar pairings of the complex moieties with a backbone hydrogen of Gly183, side group oxygens of Asp48 and Asp265, a side group hydrogen of Gln266, a side group hydrogen of Lys120, a backbone hydrogen of Arg221, a side group hydrogen of Arg221 and a side group hydrogen of Cys215 (see Figure 3B). As we may note, the theory predicts that several amino acids play similar roles in binding VO(3hp)_2_ and VO(alx)_2_. At the same time, due to the larger size of the latter complex, interactions with VO(alx)_2_ involve more distant amino acids (from the reaction centre) like Asp48 and Asp265.

Overall, Figure 3A,B present rich patterns of polar interactions for the metal complexes. Here, we note the particular features of the bonding angles involving the V atom. The large blue and cyan dots in Figure 2A1 indicate smaller O_3_VO’_4_ and O’_3_VO_4_ angles in the complexes bound to the protein compared to those observed in vacuum (small dots) and in water clusters (blue trajectory). Consistently, Figure 3A,B present structures where the ligand planes are under the estimated smaller angles. It is important to note that the theory predicts a role for the reaction centre helix Amide 1 NH group and protonated Cys215 interactions with the apical O atom of the vanadyl moieties (see dark red dotted lines in Figure 3A,B).

In the case of Zn(mal)_2_, binding is dominated by polar interactions with a side group oxygen of a Thr167, side group hydrogens of Lys128 and Arg130 and backbone hydrogens of Ala126 and Gly129 (see Figure 3C). In the case of Zn(alx)_2_, binding is dominated by polar interactions with a side group hydrogen of Lys128 and backbone hydrogens of Gly129, Arg130 and Thr131 (see Figure 3D). Compared to the V systems, Figure 3C,D indicate rich patterns of polar interactions. As a result, the structural arrangements are different due to the significantly more flexible nature of the Zn complex. Figure 3C shows Zn(mal)_2_, where the Angle1 O’_3_VO_4_ is quite large, giving rise to a nearly flat structure in the plane, resembling a distorted square pyramid. The Zn(alx)_2_ complex, in contrast, forms a nearly tetrahedral geometry (see Figure 3D).

In summary, the results of the QM/MM optimizations suggest that inductive and hydrophobic mechanisms may play a significant role in the fine fitting of long pentyl moieties into suitable protein cavities contributing to structural and electronic perturbations. This may be reflected in UV-VIS optical electronic properties.

### 2.3. Nature of Coordination Bonds

As BOMD calculations indicate similar structural and electronic tendencies in the V and Zn systems, it is thus important to address the nature of bonding in the complexes. According to the NBO analysis, in vacuum, DFT calculations predict about the same degree of ionicity in the V(12%)-O(88%) and V(29%)=O(71%) bonds for the VO(3hp)_2_ (see Table 1) and VO(alx)_2_ complexes (data in the Appendix A). Upon protein association, the ionicity of carbonyl/carboxylate bonding slightly decreases, while the V=O bond becomes more ionic. These tendencies are more pronounced for VO(alx)_2_ (see Table 1 and the images in Appendix A). Slight redistributions of the bond nature, as computed by NBO analysis, agree with the changes in the ESP atomic charges estimated according to the DFT population analysis. We may ascribe the changes in the nature of the ligand bonding to the nonpolar interactions of the long pentyl groups with the enzyme moieties and to the polar attraction of the apical O atom to the Cys215 of the PTP1B enzyme.

In comparison to bonding in oxovanadium systems, the attachment of carbonyl and carboxylate O atoms to Zn^2+^ in both complexes, Zn(mal)_2_ and Zn(alx)_2_, is significantly more ionic: Zn(5%)-O(95%) (see Table 2 and data in the Appendix A). While d-orbitals play a key role in the bonding of oxovanadium complexes, the integrity of the Zn systems is governed by sigma-type sp-electronic overlaps. The effects of both oxidation states and principal quantum numbers (nuclear charges) for the metal ions deliver comparable ionic radii and bond lengths in the two systems, but the relative ionicities of the bonds are different, as we have computed and described. As a result, the main-frame structural resemblance allows oxovanadium and Zn complexes to demonstrate parallel pharmacologic properties. Nonetheless, as we have already mentioned, the latter is expected to show better structural flexibility. Here, we may add that the larger ionicity for Zn^2+^ bonding should contribute to stronger electrostatic interactions with moieties of a hosting protein, and this may stimulate electrophilic/nucleophilic interactions with charged protein moieties promoting post-binding complex decomposition and speciation [19,20].

### 2.4. Normal Mode Analysis, VO(3hp)_2_

In Figure 4, we compare the IR and Raman spectra of the considered complexes in vacuum and in the protein environment. The spectra highlight the effects of the side groups of the pyrone moieties, the nature of the bonding to the metal cations and the structural constraints imposed by the protein. We start the normal mode analysis considering VO(3hp)_2_ (see Figure 4A1), since this complex contains the simplest ligands, namely pyromeconic acids (the Appendix A includes normal mode analysis for the neutral and dehydrated anion of 3-hydroxy-4-pyrone). According to Figure 4A1, C-O and C-C stretching of different delocalizations and phases dominate in the mid-IR spectral region, for example, the IR active mode 57 and Raman active mode 58 (at 1566 cm^−1^ and 1577 cm^−1^, respectively). At lower frequencies, the aromatic CH in-plane bending modes 40 and 41 (at 997 cm^−1^) and the V=O stretching mode 42 (at 1058 cm^−1^) present distinct resonances, which can be helpful to identify this molecule experimentally. In respect to the latter, we refer to the first record of studies of vanadyl IR activities [57]. Aromatic CH stretching modes dominate the spectra at the highest frequencies (above 3100 cm^−1^). The Appendix A provides a detailed description of prominent normal modes and includes an archive with a corresponding Gaussian output file with normal mode analysis.

In the protein environment, six additional low-frequency modes occur, namely, three translations and three rotations of the whole complex with respect to the protein cavity. Consequently, for example, mode 48 = 42 + 6 in Figure 4B1 presents V=O stretching in the complex when in the protein environment. Comparing the spectra of Figure 4A1,B1, we note that, in general, normal modes experience blue shifts upon association with the protein. Furthermore, a change in the relative IR intensities of the modes 63 and 64 (versus the corresponding modes 57 and 58), as well as degeneracy lifting at 3150 cm^−1^ in Figure 4B1, suggests a loss of symmetry in the complex due to distortion upon association with the protein (see description of the structural properties in the previous section). These modes may provide valuable experimental markers of the presence and the structural state of VO(3hp)_2_.

### 2.5. Normal Mode Analysis, Zn(mal)_2_

Maltol is the next structural augmentation of 3-hydroxy-4-pyrone, where methyl is the R_2_ residue. In Figure 4A2,B2, we display the IR and Raman spectra computed for the Zn(mal)_2_·H_2_O complex in vacuum and the Zn(mal)_2_ complex in the protein environment, respectively. In the Appendix A, we provide a detailed description of the normal modes of Zn(mal)_2_·H_2_O, as well as the spectra computed for the Zn(mal)_2_ and Zn(mal)_2_·2H_2_O complexes under tetrahedral and octahedral geometries, respectively. Here, we may note that the aromatic CH in-plane bending modes 40 and 41 and V=O stretching mode 42 present distinct resonances, which can be helpful to identify this molecule experimentally. In the protein environment, the normal modes experience a blue shift (comparing the spectra in Figure 4A1,B1). Furthermore, a change in the relative IR intensities of modes 63 and 64 (versus the corresponding modes 57 and 58), as well as the degeneracy lifting at 3150 cm^−1^ in Figure 4B1, suggests a loss of symmetry in the complex due to distortion upon association with the protein (see description of the structural properties in the previous section).

Considering the results of previous theoretical and experimental (in KBr powder) studies of the normal modes of maltol [58] and Zn(mal)_2_·H_2_O [59], in the Appendix A, we present computed spectra and image displacements of the representative normal modes of deprotonated and protonated maltol, as well as of Zn(mal)_2_·H_2_O. Accordingly, we may ascribe the IR transitions observed at 852, 922, 1202, 1277, 1459, 1513, 1577 and 1610 cm^−1^ to the 42, 45, 56, 60, 68, 73, 75 and 77 + 78 modes, respectively. Consistently, we may ascribe Raman transitions detected at 539, 718, 1045, 1366, 1465, 1511 and 1602 cm^−1^ to the 31, 38 + 39, 53, 62, 67, 72 and 78 modes, respectively. The Appendix A includes an archive with a corresponding Gaussian output file with normal mode analysis.

In the protein environment, the normal modes of Zn(mal)_2_ demonstrate both a shift to the blue as well as a stronger dispersion of frequencies, as seen in Figure 4A2,B2. These are the signatures of structural distortion and symmetry lowering. Our results indicate that modes 74 (at 1566 cm^−1^) and 76 (at 1611 cm^−1^) of Zn(mal)_2_ in PTEN may provide valuable experimental markers of the presence and the structural state of Zn(mal)_2_.

### 2.6. Normal Mode Analysis: VO(alx)_2_

Let us consider now the vibrational properties of the VO(alx)_2_ complex, where the aromatic CH moieties are replaced with methoxy and extended hydrocarbon groups, which should allow for a more specific structural attuning of the complex and the protein cavity upon association. The Raman intense mode 95, at 1050 cm^−1^ (Figure 4C1), is due to V=O stretching. Given its dominant role in the spectral region around 1100 cm^−1^, this mode may serve as a helpful spectral marker to track the presence of such a complex. The same role could be played by the intense IR active modes 159 and 160 (at 1507 and 1508 cm^−1^), which are C=O antisymmetric and symmetric stretchings, respectively. The Appendix A provides a detailed description of prominent normal modes and includes an archive with a corresponding Gaussian output file with normal mode analysis.

In the protein environment (see Figure 4D1), normal mode 95 + 6 = 101 (at 1068 cm^−1^) is specific to V=O stretching, but not unique if we compare it with the case of VO(3hp)_2_. Specifically, in the case of VO(alx)_2_ in PTP1B, the theory computes a mixing of V=O stretching with delocalized CH_2_ bendings; there is another intense transition denoted as 102 in Figure 4D1 (1070 cm^−1^). Further, for the complex in the protein environment, there are two intense IR modes, 166 (at 1519 cm^−1^) and 168 (at 1548.2 cm^−1^), as well as a strong Raman mode 170 (at 1587 cm^−1^) localized on the same ligand. The results of our theoretical studies suggest a great sensitivity of the complex’s vibrational properties to the anisotropy of interactions within the embedding cavity. We believe the provided description of the nature of the most intense normal modes may allow a helpful diagnostic to address the binding process when using FTIR and Raman spectroscopy.

### 2.7. Normal Mode Analysis: Zn(alx)_2_

Considering the rather effective association of a water molecule (under apical geometry) predicted for the Zn complexes in water clusters using BOMD, first, we computed the vibrational properties of Zn(alx)_2_·H_2_O in vacuum, where the water was under apical geometry. For this complex, dominant in the mid-infrared are IR active CH bending mode 109 (at 1136 cm^−1^), IR active delocalized C-O and C-C stretching vibrations 163 and 165 (at 1512 cm^−1^ and 1550 cm^−1^, respectively) and the Raman active delocalized C-C stretching mode 167 (1584 cm^−1^), which may all prove to be suitable for experimental diagnostics.

In the protein environment, as for the other considered systems, the theory predicts blue shifts for the Zn(alx)_2_·H_2_O normal modes. Promising for diagnostics, a nearly degenerated doublet of IR transitions 106 and 107 that involves CH bendings of the methoxy and methyl group peaks at 1146 cm^−1^; IR active transitions 161 and 164 (at 1520 cm^−1^ and 1557 cm^−1^, respectively) are due to C-C and C-O stretching delocalizations; and a prominent peak at 1594 cm^−1^ (mode 167) presents Raman expression for a C-C and C-O stretching delocalization. Overall, the computed vibrations of the complex in PTEN demonstrate wider spectral distributions of the resonances and stronger localization tendencies, reflecting lower symmetry due to structural distortions resulting from hindered interactions with the protein.

### 2.8. Oxovanadium System UV-VIS Properties

In Figure 5A,B, we report the electronic absorption and circular dichroism (CD) spectra computed for VO(3hp)_2_ in vacuum and in the PTP1B protein environment. First, a significant red shift (about 10 nm) for the HOMO-LUMO and the next electronic transitions upon protein–complex association is observed: this shift is more than 100 nm for the HOMO-LUMO. The NTOs for the HOMO-LUMO transition of the complex in vacuum and in the protein environment, reported in Figure 6, suggest a similar electronic pattern, in which the d_xy_ to π*(d_xz_-p_x_) components (admixed with ligand p electrons) dominate. This resembles the character of the HOMO-LUMO transition computed for VO(acac)_2_ reported in the literature [60]. An admixture with ligand electronics is stronger for the complex that is distorted in the protein cavity. It is interesting that while the delocalized ligand electronics determine the third and fourth electronic transitions of the complex in vacuum, in the protein environment, localized ligand electronics take the lead in the fourth and fifth transitions. Calculations predict a significant perturbation of the electronic properties of the complex upon interactions with phosphatase.

Additionally, in Figure 5B, using a thin dashed blue line, we display the CD spectrum obtained for the VO(3hp)_2_ Λ enantiomer in vacuum. The two CD spectra of the complex in vacuum indicate that a solution of this complex (as a racemic mixture) should not show any optical activity. In contrast, in the protein environment, a certain optical rotation is expected due to the conserved geometry of the embedding cavity of the enzyme’s active site (note the positive rotations for the first and second transitions by the red line in Figure 5B). Our calculations suggest a role for CD spectroscopy to monitor visible-spectral-range d-d single spin electronic transitions and ultimately probe the structural–electronic tuning of oxovanadium complexes in a protein environment.

In Figure 5C,D, we present the electronic absorption and CD spectra of the VO(alx)_2_ complex in vacuum and in the PTP1B protein environment. There is an obvious similarity in the CD spectra computed for the VO(3hp)_2_ and VO(alx)_2_ complexes in vacuum. This indicates that regardless of the side groups, the relative orientations of the electronic and magnetic transition dipole vectors are quite similar. However, the optical electronic properties of the complexes are rather different when embedded in the PTP1B enzyme.

First, the shift of the HOMO-LUMO transition for the VO(alx)_2_ complex in the protein environment is larger. Second, the CD spectra are significantly different. This suggests that the extended side groups of the VO(alx)_2_ complex introduce a “fulcrum”, and that their interactions and relative co-adjustments with the neighboring protein moieties may impose additional perturbations on the structure and electronic properties of the complex when in PTP1B. Here, it is interesting to note that while calculations anticipate that both the nature of the VO(alx)_2_ transient electronic orbital components and their reordering upon association with the protein (see Figure 7) are quite similar to those predicted for VO(3hp)_2_ (see Figure 6), the optical activities of the two complexes are significantly different. Therefore, we may infer that by selecting side groups upon synthesis and tuning the protein environment (via primary sequence change or by affecting the environment to stimulate a different conformation), one may control the frontier d-d electronic and magnetic properties of the cation in such ways that can never be achieved otherwise.

### 2.9. Zinc System UV-VIS Properties

To address how optical properties would change upon binding to a protein, for the aqueous reference, we adopted Zn complexes with a single apical water molecule coordinated such that the coordination geometry would be similar to that of a square pyramidal structure—the most likely arrangement according to the BOMD predictions for water clusters. Figure 8 presents the electronic absorption and CD spectra of Zn(mal)_2_·H_2_O and Zn(alx)_2_·H_2_O in the PTEN protein environment. In contrast to the case of oxovanadium systems, where a single d-electron determines the edge optical absorption, in the Zn systems, the computed optical transitions are due to optical transitions of the ligands. In Appendix A, we display the corresponding NTOs. For Zn(mal)_2_·H_2_O and Zn(alx)_2_·H_2_O, the frontier transitions are in the UV range, below 350 nm. These transitions involve intramolecular ligand and inter-ligand excitations. Accordingly, the TDDFT for such systems predicts a similar participation of water in optical electronic transitions. For example, see NTO pairs 5 and 6 in the upper sets in Appendix A.

Upon the association of the Zn complexes with PTEN proteins, significant red shifts (larger than 150 nm) for the HOMO-LUMO, and, next to them, optical transitions (compare red and blue line spectra in Figure 8A,C), are observed. This suggests both the electronic interactions of the complexes with the protein environment and significant structural perturbation upon binding. The blue-line spectra in Figure 8B,D present CD dispersions for the selected enantiomers computed in vacuum. In the aqueous environment, where racemic mixtures are expected, one cannot observe the optical activity for such complexes. This is the same conclusion reached for the oxovanadium systems (see discussion of Figure 5B). Instead, the optical activity of the complexes in the protein environment is expected (red lines in Figure 8B,D). Since the optical activity of the helical and β-sheet structural components of the enzymes is detected below 240 nm, the relatively red-shifted optical activity (above 300 nm) predicted for the Zn systems (due to ligand electronics) may provide a valuable diagnostic to address interactions and binding. In Appendix A, we report the NTO electronic components of the considered Zn^2+^ systems in the protein environment. Binding to the protein induces a slightly stronger localization for the red edge transitions, consistent with structural distortion.

A comparison of the electronic spectra of the Zn and V complexes in vacuum and in the protein environment indicates that these complexes may undergo significant spectral alterations upon binding to enzymes. This is particularly evident and interesting in the case of the V systems, where the protein environment shifts the computed d-d transitions into the near IR. This result suggests a new opportunity in the bioengineering of novel electro-optical and biospintronic applications, using either crystallized or surface-oriented proteins with embedded complexes. Specifically, the computed optical properties suggest that using circularly polarized light, it may be possible to engineer spin-polarized electronic excitations to detect spin polarization as an electrical signal [61,62]. This is particularly attractive because of the structural diversity of protein families and the possible modifications of enzymes (by single amino acid editing), and can be assisted by artificial intelligence to secure unique cavities to tune d-electron transitions to spectral ranges not attainable so far. From this perspective, exploring docking implications for the electronics of polyoxidovanadates [47,63] and systems alike may open a new frontier in multi-spin biospintronics.

Finally, here, it is important to note that nearly a century ago, Hans Bethe introduced crystal field theory to explain colors in transition metal complexes [64]. The considered cases of tuning the d-d electronic transitions of the oxovanadium complexes may be considered as “bio-field” electronic engineering, where the anisotropy of the embedded cavity is predictable according to its primary sequence, and variable according to the secondary structure of an adopted protein or by editing the cavity through single amino acid substitutions. In the Appendix A, we describe an example of a convolutional Neural Network to train on Coulomb forces (between the atoms of the reactive centres and of the complexes) and the wavelengths of d-d electronic resonances computed using QM/MM for this purpose, such that one may anticipate changes in d-d electronics, for example, upon a single amino acid substitution that perturbs Coulomb interactions. The Appendix A includes an archive with Mathematica codes for Neural Net training and applications. While our example offers training using wavelengths of the computed d-d transitions, one may adopt relations with more general definitions of a crystal field [65].

## 3. Methods

### 3.1. Dynamics of the Complexes in Vacuum and in Local Water Clusters

To characterize the structural features of the systems under study, we carried out Born–Oppenheimer ab initio molecular dynamics simulations for the bis(3-hydroxy-4-pyronato) oxovanadium(IV) complex, VO(3hp)_2_, and the bis(maltolato) zinc(II) complex, Zn(mal)_2_, in vacuum and in water clusters made of 25 molecules using the CP2K program [66]. We performed NVT (constant volume, constant temperature, using a Nosé thermostat [67]) simulations in cubic boxes with side-lengths of about 17 Å (the length varied ±2 Å from system to system) under periodic boundary conditions. We employed the generalized gradient approximation and the Perdew–Burke–Ernzerhof (PBE) exchange–correlation functional [68,69]. The Grimme D3 approach is taken to account for dispersive interactions [70] and the dzvp-molpot double-ζ polarization basis [71] is used for the Geodecker–Teter–Hutter pseudopotentials [72]. A time-step of 0.5 fs was employed. In each case, we performed a thermalization phase under a velocity rescaling regime [73]. Typically, thermalization in the prepared systems was reached within 300 fs. Next, we conducted NVT sampling of 1 ps. The target accuracy for the self-consistent field convergence was 10^−6^ hartree. The cut-off and the relative cut-off of the grid level were set to 400 and 100 Rydberg, respectively. The energy convergence threshold was set to 10^−12^. We conducted data analysis as well as the preparation and training of the convolutional Neural Net using Mathematica 12, Wolfram Research Inc., Champaign, Illinois, USA.

### 3.2. DFT Calculations

Considering the results reported earlier [74,75], we carried out DFT calculations for selected configurations using the LANL2DZ basis set for the V^4+^ and Zn^2+^ ions and the 6–31++G(d,p) basis set for all other atoms under the B3LYP exchange–correlation functional [76], as implemented in the Gaussian 09 program [77]. The level of the adopted theory was reported by others as efficient for application to similar molecules [78]. The natural atomic orbital and natural bond orbital analysis was conducted according to Gaussian NBO Version 3.1 [79].

To compute pre-resonance Raman intensities, we employed cphf=rdfreq instruction, as implemented under Gaussian 09. The instruction allowed setting the electromagnetic perturbation wavelength, which we ascribed to 530 nm, as it was the most commonly used wavelength in experiments. The vibrational frequencies were scaled by a factor of 0.97 (according to our FTIR measurements in metal–organic systems, as we reported elsewhere [80]). We expressed IR and Raman spectra by taking convolutions with a Lorentzian function of half-width at half-maximum of 10 cm^−1^. In the Appendix A, we present the data without scaling.

To address the optical electronic properties of the considered systems, we employed time-dependent DFT (TD-DFT) using the level of the theory employed for ground-state calculations. Optical absorption and circular dichroism spectra in the ultraviolet–visible (UV-VIS) spectral range are expressed by taking convolutions with Lorentzian line-shapes of half-width at half-maximum of 50 cm^−1^. While TD-DFT provides a description of the electronic excited states, typically, the results are ambiguous for transitions between the highest occupied molecular orbital and the lowest unoccupied molecular orbital for simple molecules only. For complex species, electronic transitions may involve several molecular orbitals without a single dominant component. In order to reduce the complexity of the considered complexes, we exploited Koopmans’s theorem [81] and searched for a transformation of the density matrix [82] to consolidate electronic redistributions specific to a selected transition as the “lower” and the “upper” orbital components. Pairs of such orbital components form the so-called natural transition orbitals (NTOs) [83]. By visualizing the NTOs of the electronic transitions responsible for optical density in the visible spectral range, we obtained a systematic description of the electronic components governing spectral responses in the visible spectral range dependent on the conformation and the degree of hydration.

### 3.3. Molecular Mechanics, Docking and ONIOM Studies

We employed the NAMD program [84] to add hydrogen atoms and to confirm the structural consistency of the proteins. In this study, we adopted Cys residues protonated according to their common pK_a_, c.a. 8.5. Since we would explore the consequences of the most effective binding of oxovanadium complexes to the active site of PTP1B, to avoid strong electrostatic repulsion between the oxovanadium oxygen and the negatively charged thiol, we protonated PTP1B Cys215 residue. Here, it is important to note that commonly, it is assumed that due to its unusually low pK_a_ (between 4.5 and 5.5, compared to a typical Cys pK_a_, c.a. 8.5 [85]), deprotonated Cys215 should play the key role of a nucleophile interacting with the phosphate group. However, there is a discussion which suggests that the protonation and hydration of Cys215 may be important for Michaelis complex formation [86,87,88,89].

Docking calculations were performed using Autodock 4 software [90] under the scoring and minimization Vina setting [91]. The docked conformations of each ligand were ranked according to their binding energies. For QM/MM studies, we sorted the top-ranked structures. To refine the docked structures and to express the optical properties of the complexes when in the protein environment, we employed a two-layer ONIOM (our own *n*-layered integrated molecular orbital and molecular mechanics) approach [92]: a complex is treated at the DFT level, while a protein is held according to molecular mechanics force field.

Specifically, when using a two-layered approach, we adopted the AMBER force field [93] for the protein and DFT setting for each complex, as we described in the DFT subsection. The step from docking to ONIOM optimization accounts for the full protonation of the docked structure. This may introduce strongly hindered interactions. Thus, to allow for a soft structural relaxation, we realized the first 4 optimization cycles, allowing full flexibility to proximal (to the complex protein) moieties, while freezing the rest of the protein and the central metal cation. Consequently, we unlocked the complex and proximal side groups for optimization while keeping the rest of the protein in a frozen configuration.

## 4. Conclusions

Using Born–Oppenheimer molecular dynamics simulations [66], DFT [77] and the QM/MM [77,92] approach, we address the structural and electronic properties of insulin-mimetic oxovanadium and Zn complexes in vacuum, in water clusters and when bound to PTEN and PTP1B phosphatases. In an aqueous environment, Zn(mal)_2_ departs from its minimum-energy tetrahedral geometry to bind a water molecule softly under apical geometry. The association of water occurs via sigma p-electron interactions, which are at the expense of electronic depletion on the carbonyl and carboxylate oxygens, as well as on the metal cation. Since water association with Zn(mal)_2_ in water clusters is dynamic, computing this complex in vacuum (under DFT) cannot lead to a structure similar to that of VO(3hp)_2_. Only when using BOMD simulations in explicit water do we observe a structural change from the tetrahedral geometry of Zn(mal)_2_ to Zn(mal)_2_·H_2_O, an arrangement analogous to that of VO(3hp)_2_. In this respect, a dynamic approach to the structural and electronic properties of the complexes in water clusters, as allowed by BOMD simulations, seems to be indispensable to access the structural realizations of Zn^2+^ complexes.

Employing NBO analysis [77,79], we evaluated similar degrees of covalency/ionicity in the coordination bonding of VO(3hp)_2_ and VO(alx)_2_ complexes, as computed in vacuum. Comparatively, upon protein association, the ionicity of carbonyl/carboxylate bonding decreases, while the V=O bond becomes more ionic. In comparison, coordinating Zn^2+^ in both complexes, Zn(mal)_2_ and Zn(alx)_2_ are significantly more ionic. While d-orbitals play the key role in the bonding of oxovanadium complexes, the integrity of the zinc systems according to sigma-type sp-electronics overlaps. The effects of both oxidation states and principal quantum numbers (nucleus charges) for the metal ions deliver comparable ionic radii and bond lengths in the two systems, but the relative ionicities of the bonds are different, as we have computed and described. As a result, the main-frame structural resemblance provides oxovanadium and zinc complexes that demonstrate parallel pharmacological properties. At the same time, the results of theoretical studies predict the zinc complexes to demonstrate better structural flexibility, which may lead to different post-binding speciation processes.

Using a docking protocol [90,91] and a two-layer ONIOM (our own n-layered integrated molecular orbital and molecular mechanics) approach [92], we modelled and refined, respectively, the embedding of the considered complexes in a protein environment to occupy space in the reactive centres. Accordingly, we described possible polar interactions to provide effective binding to the two protein cavities. For both the zinc and oxovanadium systems, the theory predicts possible effective interactions with the reaction centre helix Amide 1 NH group and proximal residues.

Assisted with normal mode analysis, we characterized vibrations that may be suitable for the experimental verification of the structural states of the considered complexes, as well as the efficiency of binding. TDDFT studies indicate significant alterations in the optical electronic properties of the complexes upon association with the enzymes. This is particularly evident and interesting in the case of the vanadium systems, where the protein environment shifts the computed d-d transitions into the near-infrared. In contrast to the case of the oxovanadium systems, the optical transitions computed for the Zn^2+^ molecular systems are due to the optical transitions of the ligands. Considering the results of our theoretical studies, we discussed the engineering of AI-assisted protein embedding to alter the electronic states of metal centres, which may be beneficial for biomedical and quantum information applications for the biospintronics of tomorrow.

## Figures and Tables

**Figure 1 molecules-30-01469-f001:**
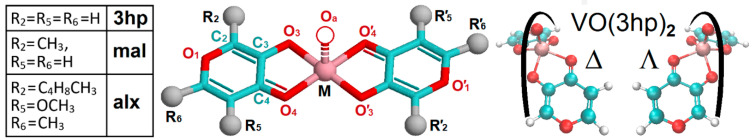
Structural layout of the considered Zn and oxovanadium complexes. The apical oxygen atom O_a_ is absent in the case of Zn systems computed in vacuum. In aqueous environment, theory anticipates departure of the Zn complex from its tetrahedral geometry, and association of a water molecule with oxygen in the O_a_ position. Considered cases include: R_2_ = R_5_ = R_6_ =H for pyromeconic acids (or 3-hydroxy-4-pyrone) to form, for example, bis(3-hydroxy-4-pyronato)oxovanadium, VO(3hp)_2_; R_2_ = CH_3_ for larixinic acid (or maltol, or 3-hydroxy-2-methyl-4-pyrone) to form, for example, bis(maltolato)zinc(II), Zn(mal)_2_; R_2_ = C_4_H_8_CH_3_, R_5_ = OCH_3_ and R_6_ = CH_3_ for allixin ligand (or 3-hydroxy-5-methoxy-6-methyl-2-pentyl-4-pyrone) to bring under attention, for example, bis(allixinato)oxovanadium(IV), VO(alx)_2_.

**Figure 2 molecules-30-01469-f002:**
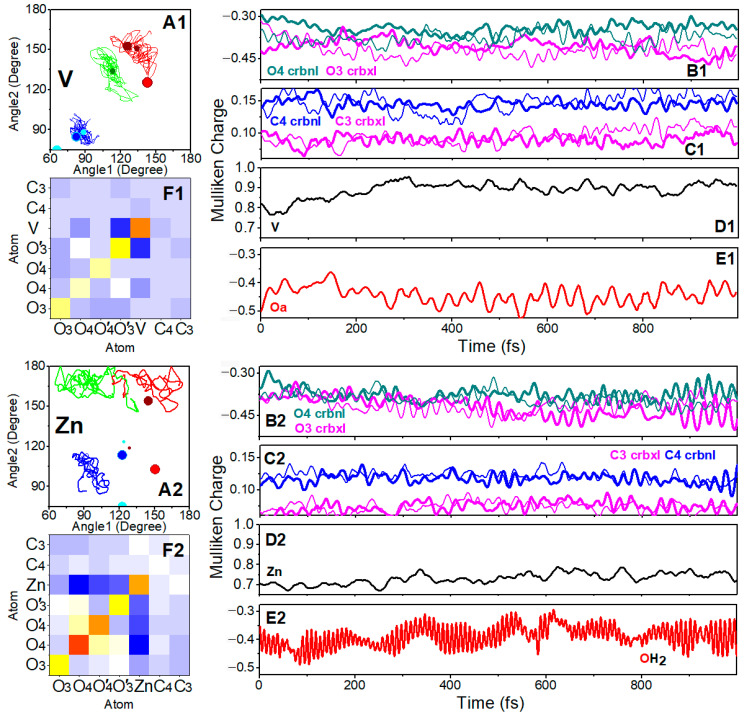
Dynamic properties of VO(3hp)_2_ and Zn(mal)_2_ complexes. (**A1**) Angle2 versus Angle1 variance along the trajectory for O_4_VO’_4_ versus O_3_VO’_3_ (red line), O_3_VO’_4_ versus O’_3_VO_4_ (blue line) and O’_3_VO_3_ versus O_a_VO_3_ (green line) for VO(3hp)_2_ in water clusters, computed using BOMD. Small dark red, cyan and dark green points indicate angular realizations computed for the complex in vacuum using DFT. Large dark red and cyan points indicate angular realizations computed for the VO(3hp)_2_ complex at PTP1B using QM/MM. Large red and blue points indicate angular realizations computed for the VO(alx)_2_ complex at PTP1B using QM/MM. (**B1**–**E1**) Mulliken charges for O_4_ (dark cyan) and O_3_ (magenta) atoms, for C_4_ (blue) and C_3_ (magenta) atoms, for Vanadium cation and for O_a_ atom, respectively, as computed for VO(3hp)_2_ in water clusters using BOMD. Thick and thin lines represent results for the atoms that belong to different ligands. (**F1**) Covariance matrix for Mulliken charges for VO(3hp)_2_ atoms. Panels (**A2**–**F2**) present analogous data for Zn(mal)_2_. Panel (**E2**) presents the Mulliken charge for oxygen atom of the water molecule nearest to Zn^2+^. Large dark red and cyan points indicate angular realizations computed for the Zn(mal)_2_ complex at PTEN using QM/MM. Large red and blue points indicate angular realizations computed for the Zn(alx)_2_ complex at PTP1B using QM/MM.

**Figure 3 molecules-30-01469-f003:**
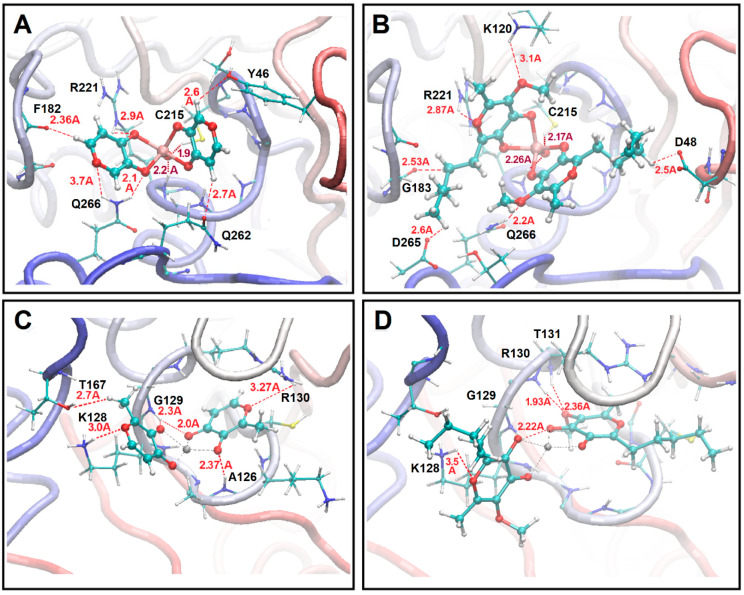
Ball and stick representations of the structural arrangements of the complex–enzyme systems obtained by docking realizations followed by ONIOM QM/MM optimizations. (**A**,**B**) VO(3hp)_2_ and VO(alx)_2_ at the PTP1B reactive site, respectively. (**C**,**D**) Zn(mal)_2_ and Zn(alx)_2_ at PTEN reactive site, respectively. For the sake of clarity, only the protein residues interacting with the complexes are reported in the diagrams. Tubular representation is adopted for the other parts of the protein.

**Figure 4 molecules-30-01469-f004:**
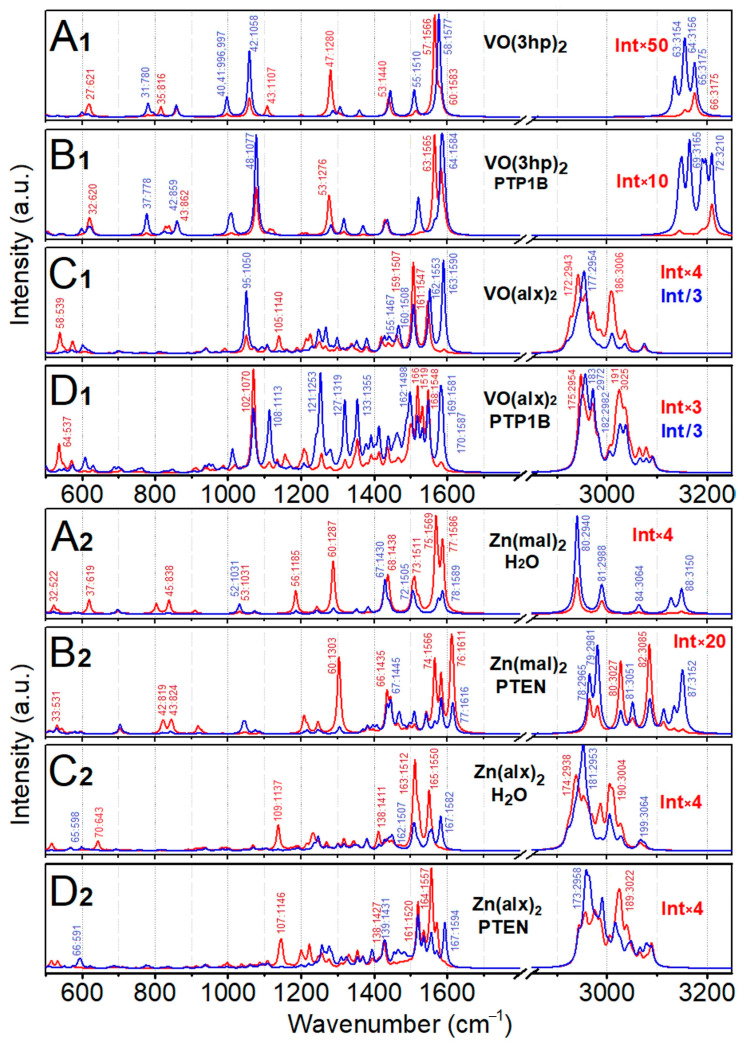
IR (red line) and Raman (blue line) spectra computed for the following: VO(3hp)_2_ in vacuum (**A1**), VO(3hp)_2_ in the PTP1B protein environment (**B1**), VO(alx)_2_ in vacuum (**C1**), VO(alx)_2_ in the PTP1B protein environment (**D1**), Zn(mal)_2_·H_2_O in vacuum (**A2**), Zn(mal)_2_ in the PTEN protein environment (**B2**), Zn(alx)_2_·H_2_O in vacuum (**C2**) and Zn(alx)_2_ in the PTEN protein environment (**D2**). The spectra are normalized, while frequencies are scaled by the factor 0.97. The Raman spectra are computed accounting for the excitation wavelength at 532 nm. All bands in the spectra are convoluted with a Lorentzian function of 10 cm^−1^ half-width at half-maximum. The resonances of OH stretching modes are out of the adopted spectral range.

**Figure 5 molecules-30-01469-f005:**
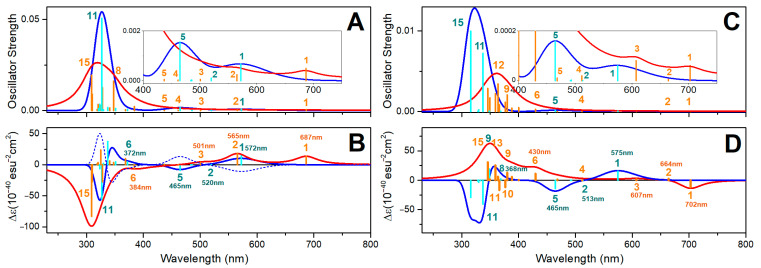
Optical electronic properties of the VO(3hp)_2_ and VO(alx)_2_ complexes. (**A**) Optical absorption spectra computed for VO(3hp)_2_ in vacuum (blue line) and in the PTP1B protein environment (red line); cyan and orange vertical lines specify the contributions of the single electronic transitions. (**B**) Optical CD spectra computed for VO(3hp)_2_ in vacuum (blue line) and in the PTP1B protein environment (red line); cyan and orange vertical lines specify the contributions of the single electronic transitions. Dashed blue line represents the optical CD spectrum of the VO(3hp)_2_ Λ enantiomer; if in a racemic mixture, the system should not demonstrate optical rotation. (**C**,**D**) show the same data computed for the VO(alx)_2_ complex in vacuum and in the PTP1B protein environment. We indicate the wavelengths of the transitions, where d electronics dominate.

**Figure 6 molecules-30-01469-f006:**
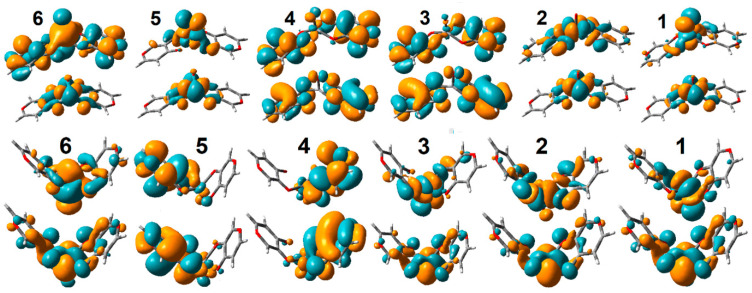
Selected NTO pairs for VO(3hp)_2_ in vacuum (**top**) and in the PTP1B protein environment (**bottom**).

**Figure 7 molecules-30-01469-f007:**
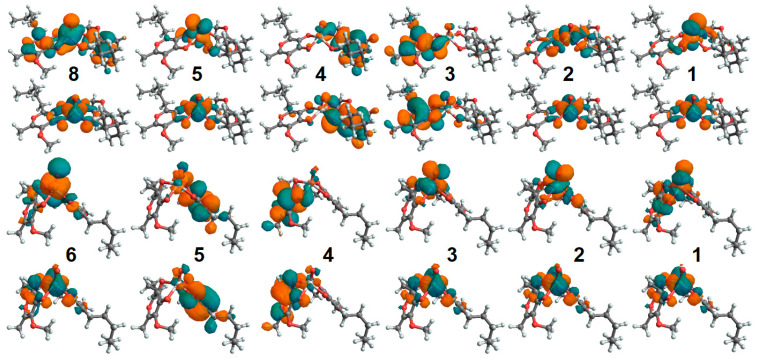
Selected NTO pairs for VO(alx)_2_ in vacuum (**top**) and in the PTP1B protein environment (**bottom**).

**Figure 8 molecules-30-01469-f008:**
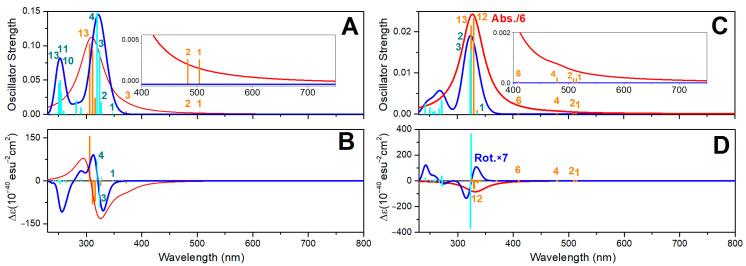
Optical electronic properties of Zn(mal)_2_ and Zn(alx)_2_ systems. (**A**): Optical absorption spectra computed for Zn(mal)_2_·H_2_O (blue line) in vacuum and when associated with a PTEN protein (red line); cyan and orange lines specify contributions of resonances. (**B**): Optical rotation spectra computed for Zn(mal)_2_·H_2_O (blue line) in vacuum and for Zn(mal)_2_ associated with a PTEN protein (red line); cyan and orange lines specify contributions of resonances. If a racemic mixture, the system should not demonstrate optical rotation. (**C**,**D**) present analogous data for Zn(alx)_2_·H_2_O in vacuum and for Zn(alx)_2_ associated with a PTEN protein.

**Table 1 molecules-30-01469-t001:** Properties of VO(3hp)_2_ complex using NBO analysis. Appendix A includes corresponding data for VO(alx)_2_ complexes. Here, carbonyl and carboxylate are noted as crbn and crbx.

Bond	Length Å	ESP Charges	Oxygen Contribution		Metal Contribution		
hp: V-O_crbn_	2.05	−0.5985, −0.5997	88.27% s(23.53%)	p^3.25^(76.42%)	11.73% s(19.76%)	p^0.96^(18.91%)	d^3.10^(61.33%)
hp: V-O_crbx_	1.97	−0.5478, −0.5493	87.66% s(26.00%)	p^2.84^(73.95%)	12.34% s(21.40%)	p^1.39^(29.77%)	d^2.28^(48.83%)
hp: V=O	1.5790	V: 1.4057 O: −0.5655	71.49% s(18.16%)	p^4.50^(81.74%)	28.51% s(13.17%)	p^0.00^(0.04%)	d^6.59^(86.79%)
			77.11%	p^1.00^(99.86%)	22.89%	p^1.00^(32.16%)	d^2.11^(67.84%)
P-hp: V-O_crbn_	2.04, 207	−0.5923, −0.5681	88.10% s(23.74%)	p^3.21^(76.21%)	11.90% s(18.66%)	p^1.40^(26.49%)	d^2.79^(54.14%)
P-hp: V-O_crbx_	1.97, 1.96	−0.5183, −0.5494	86.12% s(25.27%)	p^2.95^(74.68%)	13.88% s(20.64%)	p^1.25^(25.81%)	d^2.59^(53.55%)
P-hp: V=O	1.57968	V: 1.292 O_a_: −0.6463	72.92% s(21.05%)	p^3.75^(78.88%)	27.08% s(18.98%)	p^0.01^(0.23%)	d^4.26^(80.78%)
			77.49%	p^1.00^ (99.85%)	22.51%	p^1.00^ (30.53%)	d^2.10^(69.39%)

**Table 2 molecules-30-01469-t002:** Properties of Zn(mal)_2_ and Zn(mal)_2_·H_2_O complexes using NBO analysis. Appendix A includes corresponding data for Zn(alx)_2_ complexes.

Bond	Length Å	ESP Charges	Oxygen Contribution	Metal Contribution
mal: Zn-O_crbn_	2.07	−0.72; −0.74 Zn: 1.441	95.62% s(14.83%) p^5.74^(85.10%)	4.38% s(22.66%) p^3.39^(76.73%) d^0.03^(0.62%)
mal: Zn-O_crbx_	1.99		94.81% s(16.71%) p^4.98^(83.23%)	5.19% s(27.22%) p^2.65^(72.08%) d^0.03^(0.70%)
mal: Zn-O7_crbn_	2.10	O7;18: −0.71; −0.65 O9;11: −0.57;−0.67 Zn: 1.25 Oa: −0.81	96.18% s(15.58%) p^5.41^(84.36%)	3.82% s(20.82%) p^3.14^(65.29%) d^0.67^(13.89%)
mal: Zn-O18_crbn_	2.08		95.97% s(16.29%) p^5.13^(83.65%)	4.03% s(22.66%) p^2.88^(65.30%) d^0.53^(12.04%)
mal: Zn-O9_crbx_	2.03		96.03% s(17.82%) p^4.61^(82.13%)	3.97% s(23.36%) p^2.13^(49.82%) d^1.15^(26.81%)
mal: Zn-O11_crbx_	2.08		96.67% s(16.97%) p^4.89^(82.98%)	3.33% s(19.41%) p^2.67^(51.81%) d^1.48^(28.78%)
mal: Zn-O_a_	2.23		97.64% s(30.23%) p^2.31^(69.72%)	2.36% s(13.77%) p^4.97^(68.48%) d^1.29^(17.75%)
P-mal: Zn-O7_crbn_	2.02121	O7;O16:−0.69;−0.82 O9; O23:−0.67;−0.64 Zn 1.34	95.62% s(14.05%) p^6.11^(85.88%)	4.38% s(25.69%) p^2.83^(72.61%) d^0.07^(1.70%)
P-mal: Zn-O16_crbn_	2.24524		97.40% s(8.40%) p^10.90^(91.53%)	2.60% s(14.56%) p^5.72^(83.32%) d^0.15^(2.12%)
P-mal: Zn-O9_crbx_	2.05197		96.16% s(12.15%) p^7.22^(87.79%)	3.84% s(25.85%) p^2.79^(72.08%) d^0.08^(2.06%)
P-mal: Zn-O23_crbx_	1.94144		95.21% s(14.87%) p^5.72^(85.06%)	4.79% s(33.55%) p^1.92^(64.55%) d^0.06^(1.90%)

## Data Availability

The data presented in this study are available from the authors on reasonable request.

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
