# Peer review of "Binding Zinc and Oxo-Vanadium Insulin-Mimetic Complexes to Phosphatase Enzymes: Structure, Electronics and Implications"

_molecules, 2025, doi:10.3390/molecules30071469_

Round 1
Reviewer 1 Report
Comments and Suggestions for Authors
The article under the title “Zinc and oxo-Vanadium complexes upon binding to phosphatase 1 enzymes: structure, electronics and implications” by Volkov and coworkers presents a comprehensive theoretical examination of the structural and protein binding properties of selected complexes. The article is well-written, and the applied experimental and theoretical methods are suitable for this type of system. The article could be of potential interest to the readers of the Molecules, although there are some points that should be addressed before the final decision. Therefore, my recommendation is MAJOR REVISION.
The authors should answer the following:
- “Borrowing from this summary” in the abstract is not an appropriate term
- Vanadium (V) should be written without empty space
- The authors should give clear reasoning for selecting specific ligands in complexes, as this is not clear from the introduction
- The authors should specify why the same ligands were not used for both metal centers
- The authors should verify that the selected level of theory was appropriate for the investigated systems by citing appropriate references
- Based on the results incorporating water molecules, is it possible to conclude that complexes undergo structural changes upon dissolution?
- The authors should mention specific amino acids that are included in the protein binding. How do ligands influence the interactions with amino acids?
- The authors should compare the theoretical vibrational spectra with experimental, if available.
- The abstract should be rewritten to reflect better what was done in the manuscript, especially the last sentence does not cover material that is given in the manuscript.
Reviewer 2 Report
Comments and Suggestions for Authors
In their manuscript “Zinc and oxo-Vanadium complexes upon binding to phosphatase 1
enzymes: structure, electronics and implications“ Volkov, Perry, and Chelli present computations of oxovanadium and zinc complexes in vacuum, in water, and in two phosphatase proteins.
As anticipated, the environment has substantial impact on the structural and electronic properties of the metal complexes, with changes in the coordination geometry of, in particular, Zn, that results in more similar than different structures of the two metal complexes in aqueous or protein environment. In contrast, comparison of the impact of the protein environment on the electronic structure of the Zn or V complexes reveals Zn becoming more ionic.
Computation of IR, Raman, and UV/Vis spectra provides a means to compare with experiment and hence probe and verify the structure of the studied system.
The manuscript constitutes a significant body of work, with many data, and valuable insight into environmental effects on transition metal complexes.
There are only a few points that require attention before publication can be recommended:
The authors claim to have started the BOMD simulations from “global minimum structures … in vacuum obtained by DFG”. Are these indeed global minima? How were they obtained and verified? What do other, local minima look like and how do they differ in energy?
The attraction of one, or more, water molecules by a Zn-ion in aqueous environment, but mot only there, and hence deviation from a tetrahedral geometry, is actually not an unknown phenomenon [see e.g. J. Phys. Chem. B 2006, 110, 4, 1889–1895].
In the ONIOM calculations, which part of the system is treated with DFT and which with Amber?
The numbering of the modes in the computed IR and Raman spectra is one possibility to be able to refer to them in the text. Personally, I’d prefer labelling of important modes by the wavenumber (integer numbers only). This means more text in the figures, but allows to immediately suspect a correspondence between e.g. modes 75 and 63 (in A1 and B1).
The description of the computed IR and Raman spectra is rather lengthy. Perhaps the authors can restrict this to the most important features and refer to the supplementary material for all other assignments.
Fig 2 F1 and F2: It would be easier to read when labelling the axis with atom names instead of numbers
Fig 2 A2 is explicitly labelled with “Zn”. The same should be done for “V” in Figure 2 A1.
Round 2
Reviewer 1 Report
Comments and Suggestions for Authors
The authors answered all of the questions properly. The manuscript is suitable for publication in the present form.